# Evaluation of the Biostimulant Activity and Verticillium Wilt Protection of an Onion Extract in Olive Crops (*Olea europaea*)

**DOI:** 10.3390/plants13172499

**Published:** 2024-09-06

**Authors:** Ana Falcón-Piñeiro, Javier Zaguirre-Martínez, Ana Claudia Ibáñez-Hernández, Enrique Guillamón, Kristell Santander, Belén Barrero-Domínguez, Silvia López-Feria, Dolores Garrido, Alberto Baños

**Affiliations:** 1DMC Research Center, Camino de Jayena 82, 18620 Alhendín, Granada, Spain; anafalcon@dmcrc.com (A.F.-P.);; 2Department of Plant Physiology, University of Granada, Fuentenueva s/n, 18071 Granada, Granada, Spaindgarrido@ugr.es (D.G.); 3Neval, Camí del Horts s/n, 12592 Xilxes, Castellón, Spain; 4Dcoop Sociedad Cooperativa Andaluza, Carretera Córdoba s/n, 29200 Antequera, Málaga, Spain

**Keywords:** olive tree, onion extract, *Allium cepa*, biostimulant, antifungal, *Verticillium dahliae*

## Abstract

The olive tree is crucial to the Mediterranean agricultural economy but faces significant threats from climate change and soil-borne pathogens like *Verticillium dahliae*. This study assesses the dual role of an onion extract formulation, rich in organosulfur compounds, as both biostimulant and antifungal agent. Research was conducted across three settings: a controlled climatic chamber with non-stressed olive trees; an experimental farm with olive trees under abiotic stress; and two commercial olive orchards affected by *V. dahliae*. Results showed that in the climatic chamber, onion extract significantly reduced MDA levels in olive leaves, with a more pronounced reduction observed when the extract was applied by irrigation compared to foliar spray. The treatment also increased root length by up to 37.1% compared to controls. In field trials, irrigation with onion extract increased the number of new shoots by 148% and the length of shoots by 53.5%. In commercial orchards, treated trees exhibited reduced MDA levels, lower *V. dahliae* density, and a 26.7% increase in fruit fat content. These findings suggest that the onion extract effectively reduces oxidative stress and pathogen colonization, while enhancing plant development and fruit fat content. This supports the use of the onion extract formulation as a promising, sustainable alternative to chemical treatments for improving olive crop resilience.

## 1. Introduction

The olive tree (*Olea europaea*) is one of the most economically important crops in the Mediterranean region [1]. In 2017, the total area devoted to olive cultivation worldwide was approximately 10.8 million ha and olive production was 2,087,278 tons [2]. While countries of the Mediterranean basin are responsible for more than 90% of the world’s olive production [3], olive oil and table olive consumption extend to 179 countries [4]. The localized production together with the high global demand for olive products underlines not only the economic, but also the social and environmental importance of this crop in producing countries. Despite being one of the crops best adapted to the climate of the Mediterranean region [5], characterized by warm and dry summers, with high levels of solar radiation [6], climate change has a detrimental effect on olive performance [7]. This area is highly vulnerable to adverse environmental conditions caused by climate change [8]. According to the European Commission’s Joint Research Centre (JRC), the major effects are extremely high temperatures and changes in precipitation patterns, with the latter becoming ever scarcer, leading to severe drought events [9]. Temperature and water availability have been reported to be the main factors that compromise olive growth and yield, respectively [10]. Additionally, climate change may be related to an increased incidence of microbial diseases, such as Verticillium wilt caused by *Verticillium dahliae* [11]. 

*V. dahliae* is an ascomycete fungus responsible for the most relevant soil-borne disease affecting Mediterranean olive crops [12]. The incidence of Verticillium wilt is highly influenced by the population of the fungus in the soil, which is significantly affected by soil temperature and humidity [13]. Moreover, *V. dahliae* forms resistant structures, called microsclerotia, that can survive up to 10 years in the soil in the absence of a host and during adverse conditions [14]. Microsclerotia germination depends on water availability and is induced in response to root exudates [15]. They form hyphae that penetrate the roots, reach the xylem vessels, and give rise to conidia. They extensively colonize the xylem, which leads to water stress and causes yield losses and tree mortality [16]. Specific symptoms rely on the infective pathotype: defoliating (D) and non-defoliating (ND) [17]. The D pathotype can be lethal and causes the drop of green leaves and complete defoliation, whereas the ND pathotype is less aggressive and is associated with a slow-decline syndrome [18]. 

Because of the general awareness regarding the abiotic and biotic stress threatening olive crops, current efforts have been directed towards the development of natural, eco-friendly solutions, such as plant-based extracts, which can enhance the resilience of crops to these stresses while reducing the reliance on synthetic chemicals [19,20]. Plant extracts are regarded as safe, low in toxicity, and highly biodegradable [21]. Their use as biostimulants and natural antimicrobials has gained significant attention in recent years, showing promising results in enhancing crop performance and disease resistance [22]. Many authors have described the application of seaweed (*Ascophyllum nodosum*), moringa (*Moringa oleifera*), and liquorice (*Glycyrrhiza glabra*) extract, among others, as biostimulants in a variety of woody fruit crops, including citrus plants (*Citrus sinensis* and *Citrus clementina*), grapes (*Vitis vinifera*), and trees from the *Prunus* genus, such as plum, peach, and almond trees [23,24,25,26,27,28]. In addition, the antimicrobial activity of plant extracts such as tea (*Camellia sinensis*), thyme (*Thymus vulgaris*), *Brassica* vegetables, cork oak (*Quercus suber*), and eucalyptus (*Eucalyptus globulus*) against a wide range of phytopathogens has been reported [29,30,31,32]. 

*Allium* plants include over 900 species, with the most studied being garlic, onion, leek, and shallot. The primary bioactive compounds in *Allium* species are organosulfur compounds (OSCs), which are produced during tissue damage as a defence mechanism. Key OSCs include S-alk(en)yl-L-cysteine sulfoxides (ACSO), precursors to thiosulfinates, thiosulfonates, and sulfides. In onion (*Allium cepa*), the main sulfur compounds are isoalliin, which converts to lachrymatory factor, methiin, and propiin, which, through the action of alliinase, lead to dipropyl thiosulfinate (PTS) and propyl-propane thiosulfonate (PTSO) [33]. Extracts from the *Allium* genus have shown promising bioactive properties including antimicrobial, repellent, and antioxidant, among others [34,35]. The literature suggests that *Allium* extracts may have a dual function in agriculture as both biostimulants and antimicrobial agents. Various studies have demonstrated the capacity of garlic (*Allium sativum*) extracts to improve the growth, yield, fruit quality, and stress tolerance of legumes (*Vicia faba* and *Phaseolus vulgaris*), eggplant (*Solanum melongena*), and pepper (*Capsicum annuum*) [24,36]. Other authors have revealed that garlic extract exhibits antifungal activity similar to that of chemical fungicides carbendazim and kanamycin B against *Colletotrichum musae* or *Lasiodiplodia theobromae* (*Musa × paradisiaca*) [30]. Additionally, we demonstrated that organosulfur compounds from onion (*Allium cepa*) reduce the incidence and severity of Verticillium wilt, as well as the density of *V. dahliae* in the soil, in trials conducted in a climatic chamber with 7-year-old olive plantlets var. Picual [37].

The aim of this study was to assess the dual biostimulant and antifungal activity of an onion extract formulation rich in organosulfur compounds. This investigation was conducted using three experimental approaches: (1) a controlled climatic chamber trial to assess the biostimulant effect on non-stressed, 1-year-old olive trees, focusing on oxidative stress and plant development parameters; (2) a field trial on an experimental farm to investigate the biostimulant effect on budbreak and shoot length in 4-year-old olive trees experiencing abiotic stress; and (3) two field trials in commercial olive orchards to evaluate both the biostimulant and antifungal effects on mature olive trees naturally infected with *V. dahliae*, by measuring oxidative stress and fruit quality parameters, as well as pathogen density. This multi-tiered approach was designed to provide comprehensive insights into the potential of the onion extract as a sustainable solution for enhancing olive tree resilience and controlling fungal pathogens across different environmental settings.

## 2. Results

### 2.1. Biostimulant Activity under Controlled Conditions

To determine ROS accumulation and membrane damage in control and treated 1-year-old olive plantlets in a climatic chamber, malondialdehyde (MDA) was measured. The results are shown in Figure 1. Figure 1a illustrates the changes in MDA levels in olive leaves in response to three treatment applications, administered 24 h before the first sampling, on day 15, and day 30 post-sampling. No significant differences were observed in MDA levels across all groups 24 h after the first application, indicating minimal oxidative stress and membrane disruption. Fifteen days after the first application, a significant reduction of MDA was observed in olive plantlets treated by foliar spray and irrigation compared to the control (*p* < 0.05). This difference was maintained in olive trees treated by irrigation on day 30 (30 days after the first application and 15 days after the second). In the case of olive plantlets treated by foliar spray, a statistical trend towards a reduction in MDA levels compared to control plants was observed (*p* < 0.1). Finally, on day 60 of the experiment (30 days after the third application), the MDA levels equalized.

The MDA levels in roots, showed in Figure 1b, did not show any significant variation in the treated plant with respect to control plants at any sampling time. In the case of olive plantlets treated by irrigation, a statistical trend towards a reduction in MDA was observed on days 15 and 30 of sampling (*p* < 0.1).

The antioxidant capacity was evaluated by measuring the reduction of ferric ions (Fe^3+^) to ferrous ions (Fe^2+^) using the FRAP (ferric-reducing antioxidant potential) method (Figure 2). Figure 2a depicts the concentration of Fe^2+^ in leaves over time. There were no significant differences across groups on the first and second days of sampling, even if the foliar spray treatment shows a slight reduction on day 15 compared to the control. A more pronounced effect is observed on day 30, since Fe^2+^ is significantly reduced in the foliar samples of plants treated by both foliar spray and irrigation (*p* < 0.05). This difference compared to the control disappeared 30 days after the last application of the treatment (day 60). 

Concentration of Fe^2+^ in roots is shown in Figure 2b. As in the leaf samples, no differences were observed between the different groups on the first day of sampling. A significant reduction in Fe^2+^ levels was observed in leaves from plants treated by irrigation compared to both the control and foliar spray groups 15 days after the first treatment application (*p* < 0.05). This trend persisted on day 30. By day 60, the Fe^2+^ levels in the irrigation group remained low (*p* < 0.05), with the control and foliar spray groups showing more variability, but without significant differences between them. 

The average leaf weight and root length of the olive plantlets were assessed on the final day of sampling. As shown in Table 1, the leaf weight of olive plantlets treated by foliar spray and irrigation did not significantly differ from each other or from the control. In contrast to leaf weight, root length results showed significant differences among groups. Root length significantly increased in plants treated by both foliar spray and irrigation compared to the control (*p* < 0.01). No significant differences were observed between the two application methods. 

### 2.2. Biostimulant Activity under Field Conditions

#### 2.2.1. Budbreak Induction in 4 Years Old Olive Trees 

The biostimulant activity of the onion extract was assessed through a budbreak study in 4-year-old olive trees under conditions of severe defoliation due to abiotic stress in the field trial. The results are shown in Figure 3. Significant differences were observed between the number of shoots in control olive trees and those treated by foliar spray (*p* < 0.01) and irrigation (*p* < 0.001) at both evaluation times (Figure 4). While the number of shoots on the olive trees treated by foliar spray and irrigation was similar after two applications of onion extract, after three applications the number of shoots significantly increased in the trees treated by irrigation compared to those treated by foliar spray (*p* < 0.01) (Figure 3a). The length of shoots did not differ significantly between control olive trees and those treated with foliar spray at any of the evaluation times. However, significant variations were found in olive trees irrigated with onion extract compared to the control after 2 (*p* < 0.05) and 3 applications (*p* < 0.001), and foliar spray-treated plants after 3 applications (*p* < 0.01) (Figure 3b). 

#### 2.2.2. Biostimulant Activity in Mature Olive Trees 

MDA content and the reduction of Fe^3+^ to Fe^2+^ were also determined in leaf samples from mature olive trees naturally infected with *V. dahliae* in commercial olive orchards located in Linares and Santaella. Results are shown in Figure 5 and Figure 6, respectively. In both locations, MDA levels in leaf samples from the control area were higher than in those from the treatment area. In Linares, although the difference was not statistically significant, there was a trend towards the reduction of MDA in the treated group (*p* < 0.1). In Santaella, leaves from treated trees showed a significant reduction in MDA compared to the control (*p* < 0.05). While the MDA content in the control groups at both locations was similar, Fe^2+^ levels differed significantly. At the Linares orchard, a significant reduction in Fe^2+^ was observed in the treated group with respect to the control (*p* < 0.01), whereas at Santaella they were similar. Additionally, proline content was measured. As shown in Figure 7, no significant differences were observed between control and treatment groups within the same orchard. Interestingly, proline content was significantly lower at the Santaella orchard.

The weight, volume, moisture, and fat content per 100 fruits was measured for both treated and untreated olive trees at farms located in Linares and Santaella. As shown in Figure 8a,b, the analyses of weight and volume revealed no significant differences between fruits from the control and treatment zones within the same farm. The results for weight and volume were correlated. Additionally, these data revealed significant differences between the two farms. Fruits from the farm in Linares were significantly smaller and had lower volumes compared to those from the farm in Santaella (*p* < 0.05). The analysis of humidity content (Figure 8c) revealed that there were no statistically significant differences between the moisture levels of fruits from treated and control trees in the Linares orchard. However, a significant reduction in moisture content was observed in the fruits from treated trees compared to control trees in the Santaella orchard (*p* < 0.001). Regarding fat content (Figure 8d), the fruits from treated olive trees in Santaella showed a statistically significant increase compared to the fruits from control trees (*p* < 0.001). While no statistically significant differences were observed in the fat content of fruits from Linares, there was a statistical trend towards an increase in fruit fat content in the treated trees compared to the control (*p* < 0.1). 

### 2.3. Antifungal Activity against V. dahliae under Field Conditions

The impact of the onion extract formulation on the infection rates and gene copy numbers of *V. dahliae* in olive trees was assessed in both locations (Table 2). In the Linares orchard, the percentage of infected trees was slightly higher in the treated group compared to the control, though these differences were not statistically significant. However, the log_10_ gene copies were significantly lower in the treated trees compared to the control group (*p* < 0.0001). A similar pattern was observed in the Santaella orchard. No significant differences in infection rate were observed between the control and treated groups, but a significant reduction of the log_10_ gene copies was observed in the treatment group compared to the control (*p* < 0.0001).

## 3. Discussion

The olive tree is increasingly susceptible to climate change effects [38]. These climatic shifts exacerbate existing challenges, notably the prevalence of *V. dahliae*. Climate change and disease pressure not only undermine the resilience of olive trees but also jeopardize the long-term viability of the olive oil industry [39]. A major concern in managing the effects of abiotic and biotic stress is the need to improve the sustainability of the agricultural industry through novel, eco-friendly strategies to control plant diseases and stimulate plant resilience [40,41]. In our study, we demonstrated the dual biostimulant and antifungal activity of an onion extract formulation rich in organosulfur compounds on olive trees under non-stressful conditions in a climatic chamber, as well as under abiotic and biotic stress in field trials.

In the experiment conducted in the climatic chamber, treatments were applied to olive trees in good condition that were not under abiotic or biotic stress. Our findings did not reveal any significant variations in the MDA content in roots. However, a reduction was observed in leaves on days 15 and 30 of sampling. MDA is a product of membrane lipid peroxidation, which occurs when lipids in cellular membranes are damaged by free radicals [42]. Multiple researchers have used MDA as an indicator of oxidative stress in olive trees [43]. Our results demonstrate not only that the onion extract formulation has no phytotoxic effects on 1-year-old olive trees, but they also suggest that it could reduce lipid peroxidation under stress conditions. Onion extract reduced the ferric-reducing capacity on days 15 and 30 of sampling. The ferric-reducing capacity in roots was only influenced by application through irrigation. FRAP is an indicator of a tissue’s ability to neutralize free radicals and reduce oxidative stress, providing an indirect measure of total antioxidant activity. The reduction in ferric-reducing ability, indicated by a lower Fe²⁺ content, is generally associated with increased oxidative stress [44]. However, in plants experiencing moderate or low oxidative stress, a high reducing capacity may not be necessary, as the production of reactive oxygen species (ROS) is also minimal. In this context, iron regulation could be modulated to prevent the accumulation of free Fe^2+^, which could generate ROS through the Fenton reaction [45]. Consequently, lower Fe^2+^ levels would indicate a more balanced redox state.

Although the onion extract had a significant effect on MDA and Fe^2+^ levels 15 days after the first and second applications, 30 days after the third application the levels were comparable to those of the control, which may be attributed to various reasons. On one hand, this could be due to the acclimatization of the olive plantlets in the climatic chamber. On the other hand, the volatility of the organosulfur compounds present in the onion extract could explain the absence of differences 30 days after the last application. In a previous study, we assessed the persistence of organosulfur compounds from onion extract in soil by HPLC–UV [34]. This study demonstrated that the compounds rapidly volatilize to varying degrees, with concentrations reducing by up to 50% 15 days after application. 

Additionally, results from the climatic chamber trial demonstrated that both foliar spray and irrigation treatments positively influenced root growth. Treated 1-year-old olive plantlets exhibited more robust root structures. Ali et al. [46] described a similar effect in eggplant seedlings treated with allelochemicals from *Allium* species applied via foliar spray. As stated by Pedranzani et al. [47], a more developed root structure enhances plant stress resilience by improving access to water and nutrients as well as hydraulic conductivity. *V. dahliae* colonizes the xylem of the plant and causes vessel occlusion, reducing water transport and leading to stomatal closure [48]. Therefore, stimulation of root development not only positively influences conditions under water or saline stress, but also enhances resistance to the symptoms of Verticillium wilt. This growth biostimulation effect is consistent with the results obtained at the Neval experimental farm. In the field trial involving 4-year-old trees, the application of onion extract provided a biostimulant effect, as evidenced by the increased budbreak rate. Our results are in accordance with those reported by other authors that obtained higher budbreak rate on 10-year-old apple trees exposed to sulfur natural compounds [49]. These results are of great relevance in the current context of a climate emergency. Temperature exerts a significant influence on the phenological phases of budbreak and flowering of olive trees. It also controls the winter dormancy and the onset of vegetative growth that begins in spring, stages that condition the beginning of new structure development. The increase in global temperatures as a result of climate change may substantially reduce the number of chill hours in winter and raise the average temperature in spring, which, according to several studies, adversely affect budbreak and flowering [50,51,52]. 

Both the determinations conducted in the climatic chamber and the budbreak study carried out on the experimental farm indicate that the application of the formulation by irrigation was more effective. Furthermore, the budbreak study highlights the importance of performing three successive applications by irrigation. The results show that the budbreak rate and shoot length significantly increased after the third irrigation application compared to the results obtained after the second. 

The Linares and Santaella orchards were naturally infected with *V. dahliae*. Onion extract reduced MDA content and, consequently, membrane lipid peroxidation in treated olive trees at both orchards. These results are consistent with those described in the climatic chamber trial, and they support the ability of the onion extract to reduce oxidative stress and membrane damage under stress conditions. Similarly, allelochemicals from *Allium* species have been shown to reduce MDA content in plants under biotic stress. Reduced MDA content has been documented in eggplants infected with *V. dahliae* when treated with an extract from garlic rich in these compounds [46]. Such effects could be explained by the antioxidant properties of organosulfur compounds from *Allium* species [53]. The mechanisms by which organosulfur compounds exert their antioxidant capacity have been described in previous research. According to Cascajosa-Lira et al., they scavenge free radicals and induce the activity of cellular antioxidant enzymes [54]. In addition, Llana-Ruiz-Cabello et al. reported that organosulfur compounds provide nonenzymatic antioxidant protection, demonstrating their ability to enhance lipid stability [55]. On the other hand, the results of FRAP analyses performed on olive plant tissue in both climatic chambers and field trials show that onion extract does not induce the reduction of Fe^3+^ to Fe^2+^. Similarly, it does not promote the accumulation of proline, which is known to confer a protective effect by neutralizing free radicals [56].

Water deficit, imposed by climate change, increased soil salinity due to irrigation with low-quality water, and high incidence of pathogens causing vessel occlusion, is a limiting factor in olive cultivation [57], and it affects fruit setting and quality [58]. Drought stress during the phases of fruit setting results in small fruits with low fat content [59]. Additionally, the oils obtained from these fruits have occasionally been found to be excessively bitter [2]. The application of onion extract was associated with an increase in the fat content of the fruits from *V. dahliae*-infected olive trees at both the Linares and Santaella orchards. As mentioned above, *V. dahliae* reduces hydraulic conductivity by colonizing the xylem [48]. The increase in fat content of the fruits from trees treated with onion extract could be explained by several mechanisms. In this study, onion extract has been shown to promote root development, which could improve the plant’s ability to absorb water and mitigate the effect of *V. dahliae* on hydraulic conductivity. Additionally, the antioxidant properties of the extract, as reported in this study, may protect tissues from oxidative stress caused by the infection, helping to stabilize metabolic processes. Lastly, the onion extract could have a direct effect on *V. dahliae* population, thereby limiting the severity of the damage it causes. This hypothesis is further reinforced by the RT-qPCR results. Quantification of *V. dahliae* by RT-qPCR in leaf samples from treated and control olive trees indicated that, while the treatment did not substantially reduce the percentage of infected trees, it consistently resulted in a significant reduction in the pathogen density in leaves, as reflected by the lower number of gene copies in both locations. These results are in accordance with those reported by our group in a previous study, where the effect of organosulfur compounds from onion on Verticillium wilt suppression was evaluated in 7-month-old olive plantlets var. Picual [37]. The plantlets were grown in soil artificially infested with *V. dahliae* in the climatic chamber under optimal conditions for Verticillium wilt development. The application of the organosulfur compounds by irrigation reduced *V. dahliae* density in the soil, vascular colonization, and symptom severity.

To our knowledge, the use of *Allium* extracts for the control of Verticillium wilt in olive trees has not been extensively studied, despite their strong antifungal activity against pathogenic fungi. Pomegranate (*Punica granatum*) and carob (*Ceratonia siliqua*) extracts have been shown to be effective in reducing disease severity when applied by irrigation [60]. The results of this study support the application of plant extracts by irrigation, as foliar application proved ineffective in controlling the disease.

The biostimulant and antifungal activity observed in our study are likely due to the dual effect of multiple bioactive compounds present in the onion extract, particularly organosulfur compounds such as PTS and PTSO. These compounds are known for their potent antifungal properties, which they exert by disrupting the cellular integrity and metabolic functions of pathogenic microorganisms such as *V. dahliae* [61]. This disruption may occur through the inhibition of essential enzymatic activities required for fungal growth and the destabilization of fungal cell membranes, ultimately reducing pathogen viability and colonization [62]. Moreover, these organosulfur compounds may also enhance the plant’s innate defence mechanisms by upregulating genes involved in antioxidant responses, thereby mitigating oxidative damage under both biotic and abiotic stress conditions [63,64]. The interplay between these compounds is particularly compelling, as they may work synergistically to protect against biotic stressors while supporting overall plant health by boosting antioxidant capacity and promoting growth under varying environmental conditions. This dual action could be the key to the effectiveness of the onion extract as both an antifungal and a biostimulant agent. 

The dual functionality of the onion extract formulation could not only effectively aid in stress mitigation, but it also aligns with the growing demand for sustainable agricultural practices. Nevertheless, since Picual is a cultivar primarily used for olive oil production [65], studying the potential impact of onion extract on olive oil is essential. The interaction between onion-derived compounds and olive oil constituents may influence key quality parameters such as oxidative stability and sensory attributes, which are pivotal to its desirability in the marketplace. A previous trial demonstrated that while the application of a biostimulant based on tropical fruit extract altered the organoleptic properties of olive oil from cultivar Racioppella, treatment with glycine–betaine had no effect [66]. Additionally, research with the Empeltre variety has shown that the incorporation of garlic into the olive paste during the malaxation process significantly increased the total phenolic content and antioxidant capacity of the resulting olive oil [67]. Further studies are needed to evaluate the potential effects of onion extract on oxidative stability, fatty acid, and phenolic compound content in olive oil.

Finally, given that the application of various active compounds with specific functions is a common practice in agronomic management, future studies should investigate the synergy between onion extract and other sustainable alternatives that have been shown to protect olives from abiotic and biotic stress and enhance productivity, such as seaweed extracts [68], salicylic acid [69], and sodium nitroprusside (SNP) [70,71].

## 4. Materials and Methods

### 4.1. Onion Extract

A formulation based on an onion extract (*Allium cepa* L. bulb extract) obtained from discarded onions was tested. The product (TROFIC^®^) was provided by DOMCA S.A.U. (Granada, Spain), and was composed by 85% of a standardized bulb onion extract containing 50% of organosulfur compounds derived from propiin, such as propyl thiosulfinates and thiosulfonates. It also included other components essential for stability, such as soy lecithin and ascorbic acid.

### 4.2. Evaluation of Biostimulant Activity in Climatic Chamber 

Sixty 1-year-old Picual olive plantlets, approximately 50 cm in height, were randomly selected and transplanted into new pots. They were divided into three experimental groups: control plants that received no treatment, plants treated by foliar spray, and plants treated by irrigation. The olive plantlets were numbered and kept in a climatic chamber under non-stressful conditions, with a photoperiod of 8 h of light and 16 h of darkness, and a maximum/minimum temperature of 25/12 °C. The product was diluted to 500 mg/L. A volume of 150 mL per plantlet was applied by irrigation. For the foliar spray treatment, the necessary volume to cover the entire aerial part was applied, which was approximately 3 mL. Three applications were performed: one at the beginning of the trial, another after 15 days, and a final application 30 days after the beginning of the trial. Destructive sampling of leaves and roots from 5 olive plantlets was performed at 4 different times: 24 h after the first application of the treatment (T1), after 15 days (T2), after 30 days (T3), and after 60 days (T4). On the days when sampling and treatment application coincided (T2 and T3), sampling was performed first and then the treatment was applied to the remaining plantlets. Leaves and roots were frozen in liquid nitrogen, pulverized using a grinder, and immediately stored at −80 °C until analysis. For each sampling time, MDA was determined, and FRAP analysis was performed. Additionally, after the final sampling, leaf biomass and root length were evaluated. 

#### 4.2.1. Measurement of MDA

The MDA content was determined following the TBARS procedure proposed by Heath and Packer with some modifications [72]. For each leaf sample, 300 mg were macerated with 20% (*w*/*v*) TCA (trichloroacetic acid) and 4% (*w*/*v*) BTH (butylated hydroxytoluene) and centrifuged at 12,000 rpm for 10 min. Supernatant was mixed with 0.5% (*w*/*v*) TBA (thiobarbituric acid), and the mixture was incubated at 95 °C for 30 min in a water bath. The samples were cooled on ice to stop the reaction, and then centrifuged at 12,000 rpm for 10 min. The absorbance of the supernatant was measured at 532 nm and 600 nm. A standard curve was generated using known concentrations of MDA, and results were expressed as nmol MDA/g sample. Each sample was analysed in triplicate, and the absorbance of each replicate was measured twice. 

The determination of MDA in roots was conducted following the same method using 30 mg of lyophilized samples. The lyophilization process was essential as the residual water and soil interfered with the accurate quantification of the metabolite.

#### 4.2.2. FRAP Assay

The ferric-reducing antioxidant power assay (FRAP) was conducted according to Benzie and Strain [73]. Some modifications were applied to the protocol. For each leaf sample, 300 mg were macerated with 80% (*v*/*v*) acetone and shaken for 15 min at 4 °C in darkness. After centrifugation at 5000 rpm at 4 °C for 15 min, FRAP reagent was added to the supernatant. Samples were incubated for 30 min at 37 °C in darkness, and absorbance was measured at 595 nm. The amount of Fe^2+^ in the samples was calculated from a standard curve of known concentrations. Results were expressed as µg Fe^2+^/mg sample. Each sample was analysed in duplicate, and absorbance was measured twice for each replicate.

The same protocol was conducted for the FRAP assay in root tissue, using 20 mg lyophilized samples.

### 4.3. Evaluation of Biostimulant Activity in Young Olive Trees in Experimental Farm

The field trial was conducted from March to June 2021, in collaboration with Neval, an agricultural R&D laboratory accredited by the Spanish Ministry of Agriculture for the execution of officially recognized tests, in an experimental farm located in Xilxes, Valencia, Spain (30N 740759.99 4406967.05 UTM WGS84). The soil was classified as loam, according to the USDA (United States Department of Agriculture). It contained 45% sand, 34% silt, and 21% clay (analysed by the Bouyoucos hydrometer method [74]), and 1.21% organic matter (measured according to the Walkley–Black procedure [74]). 

The biostimulant activity was evaluated in completely defoliated, 4-year-old olive trees of the Picual variety. They were transplanted from pots to soil with a planting frame of 3.5 × 2 m. The soil had not received any previous treatment. The state of severe defoliation of the olive trees was due to abiotic stress caused by the transplant. Treatment application started after a 1-month acclimation period. The number of shoots was quantified before the first application. The efficacy of onion extract was evaluated by foliar spray and irrigation in comparison with the control (water). The product was diluted to 500 mg/L water. Foliar spraying was carried out using a backpack sprayer and a spray volume equivalent to 700 L/ha, while for irrigation 10 L/tree was applied. Three applications were made at 21-day intervals. Three replicates were carried out, with 3 olive trees each, in a completely randomized manner. From transplantation until the end of the study, the weekly irrigation regime established by the farmer was maintained to avoid drought stress. To evaluate the efficacy of the treatment, the length of the shoots and the number of new ones were assessed 20 days after the second and third applications. 

### 4.4. Evaluation of Biostimulant and Antifungal Activity in Olive Orchards

The trial was conducted in two Picual olive orchards situated in Linares, Jaen (30 N 444908.38 4209274.77 UTM WGS84), and Santaella, Córdoba, Spain (30 N 335550.44 4153847.83 UTM WGS84), both of which are owned by DCOOP, a cooperative headquartered in Antequera (Málaga Spain) and recognized as the world’s largest producer of olive oil. The trial lasted 14 months, from April 2021 to June 2022. According to the USDA, the soil of Linares orchard was classified as sandy loam (68% sand, 18% silt, and 14% clay) and had 1.13% organic matter, while the soil of Santaella was classified as loamy sand (78% sand, 13% silt and 9% clay) and had 0.8% organic matter. These analyses were carried out following the same methods as those specified in Section 4.3. Both are intensively irrigated orchards. The olive trees in the Linares orchard (10 × 10 m planting frame) were bicentennial and were irrigated 12 h every 4 days. In contrast, the 25-year-old olive trees in Santaella orchard (7 × 7 m planting frame) received 3 h of daily irrigation from April to October. Both had a high incidence of Verticillium wilt, and symptoms of decline syndrome were observed. 

The orchards were divided into 2 areas: treatment and control, outlined in red and yellow in Figure 9 and Figure 10. The treatment was applied in spring, as established by the farmer, through the irrigation system at 5 L/ha. Application and sampling dates are described in Table 3. Three applications were made at 1-month intervals during the spring seasons of 2021 and 2022, starting in April. The following samplings were carried out on the following times: fruit sampling was carried out in November 2021; and leaf sampling in June 2022, 2 weeks after the last application.

Twenty olive trees were sampled in each area. The sampled olive trees, randomly selected, were located in the central area, avoiding olive trees on the edges. From each olive tree, 20 olives were taken by hand, 5 olives for each orientation (South, North, East and West). The fruits were collected at eye level, both from the most superficial branches and from the inner area of the olive tree. Sampling was performed looking the other way to avoid subjectivity, with the aim of preventing preferences for a specific olive ripeness or size. In total, 400 olives were obtained from each area and orchard.

Leaf samples were taken from 30 olive trees in each of the delimited areas, avoiding those on the edges. At least 100 leaves were collected from each tree, ensuring samples were taken from all orientations. Leaves were sampled randomly, selecting those at eye-level from both the outer branches and those in the inner area of the olive tree.

#### 4.4.1. Fruit Analysis

Fruits from 5 olive trees were analysed together, grouped according to plant proximity. Weight and volume of each group of 100 fruits were established and the average was calculated. Volume was measured using the Archimedes principle. Olives were placed in 500 ml of water and the displacement was measured. Water content was determined gravimetrically from 30 g of ground olives after drying in an air oven at 105 °C for 24 h, according to the UNE (Spanish Standard) 55031:1973 [75]. Fat content was measured from the dried sampled used in the moisture determination by Soxhlet method following UNE 55030:1961 guidelines [76]. The fat was separated with hexane in a Soxhlet extractor for 6 h. The solvent was removed in a rotary evaporator at 40 °C, and the remaining oil was dried in an oven at 60 °C until constant weight.

#### 4.4.2. Leaf Analysis

Leaf samples from each tree were numbered and analysed separately. Leaves were frozen in liquid nitrogen, pulverized using a grinder, and subsequently stored at −80 °C. The determination of MDA and the study of antioxidant capacity (FRAP method) were carried out following the protocols described in Section 4.2.

Proline was analysed from an amino acid extract. The amino acids were extracted from N2 pulverized material with a mix of ethanol/chloroform/water (12/5/1; *v*/*v*/*v*). Norvaline and sarcosine were used as internal standards. The extract was centrifuged at 3500× *g* during 10 min at 4 °C and the supernatant was separated into chloroform and aqueous phases by the addition of HCl 0.1 N and chloroform. The mixture was centrifuged at 3500× *g* during 5 min to separate phases; then, the aqueous phase was dried under nitrogen flow and stored at 20 °C under an inert atmosphere. The dried samples were resuspended in 0.9 mL of HCl 0.1 N, sonicated and filtered with nylon filter (0.22 μm), and suitably diluted. Combining o-phthalaldehyde (OPA) and fluorenylmethyl chloroformate (FMOC) chemistries was used for pre-column derivatization of amino acids. Chromatographic analysis was done by following the method proposed by Palma et al. with some modifications [77]. The amino acids were quantified by HPLC (Agilent 1260 Infinity, Agilent, Santa Clara, CA, USA) with an ACE 5 C18-PFP 4.6 mm 250 mm column and a fluorometer using excitation and emission wavelengths of 340 and 450 nm (0–15 min) and 260 and 325 nm (16–33 min). Amino acids were eluted at a flow rate of 1 mL/min using an elution gradient with sodium acetate buffer 25 mM pH 6.8 (A) and acetonitrile/methanol/water mix (45/45/10, *v*/*v*/*v*) (B). The gradient profile, expressed as (t [min]; %A), was: (0; 80%), (20; 40%), (24; 40%), (26; 0%), (31; 0%) and (33; 80%). Results were expressed as pg proline/g sample.

*V. dahliae* was detected and quantified in leaf samples by RT-qPCR. DNA extraction from leaves was carried out using the Plant/Fungi DNA Isolation Kit (Norgen Biotek, Thorold, ON, Canada) following the instructions provided by the manufacturer. The same kit was used to extract DNA from a *V. dahliae* pure culture (isolate V136I; provided by the Department of Crop Protection, Institute for Sustainable Agriculture, Spanish National Research Council, Córdoba, Spain). DNA was stored at −20 °C for further use. To generate a standard curve, *V. dahliae* V136I DNA was amplified by conventional PCR in a 2720 Thermal Cycler (Applied Biosystems, Singapore) using primers targeting the internal transcribed spacer (ITS) [78]: VerDITS-F 5′-CCGGTCCATCAGTCTCTCTG-3′ and VerDITS-R 5′-CACACTACATATCGCGTTTCG-3′. These primers amplify a 132-bp fragment. The amplification was carried out in a final reaction volume of 25 µL that contained 5 µL DNA, 2 µL primers 10 µM (5 µM-Fw + 5 µM-Rev), 12.5 µL AmpliTaq Gold 360 Master Mix (Thermo Fischer Scientific, Vilnius, Lithuania) and ultrapure nuclease-free water up to the reaction volume (Thermo Fischer Scientific, Bremen, Germany). Cycling conditions consisted of 5 min at 95 °C, 40 cycles of 5 s at 95 °C, 15 s at 55 °C and 30 s at 72 °C, and a final extension step of 7 min at 72 °C. The PCR product was run on a 1.5% agarose gel to verify the absence of other DNA fragments. DNA concentration was measured using Qubit 4 Fluorometer (Invitrogen, Darmstadt, Germany), and the number of copies was calculated. For absolute quantification, tenfold serial dilutions of DNA were prepared in nuclease-free water, yielding concentrations ranging from 10^8^ to 10^1^ copies/µL.

For the identification of infected trees and the quantification of *V. dahliae*, RT-qPCR was conducted on a Bio-Rad CFX Connect Real-Time PCR Detection System (Bio-Rad, Feldkirchen, Germany). The reaction volume (20 µL) contained 5 µL DNA, 2 µL primers 10 µM (5 µM-Fw + 5 µM-Rev), 10 µL iTaq Universal SYBR Green Supermix (Bio-Rad, Feldkirchen, Germany), and 3 µL ultrapure nuclease-free water. Each run included a positive control of *V. dahliae* V136I DNA and a no-template control (NTC) in which the DNA was replaced with nuclease-free water. Two simultaneous replicates were carried out for each sample. The RT-qPCR programme consisted of an initial step of denaturation for 5 min at 95 °C, 40 cycles of 5 s at 95 °C, 45 s at 63 °C, and 3 s at 77 °C. A melting curve was performed from 65 °C to 95 °C with a heating rate of 0.5 °C/s [79]. The initial copy number of each sample was calculated based on the slope and intercept generated by the corresponding standard curve using the RT-qPCR CFX Manager software v.3.1 (Bio-Rad). Results were expressed as Log_10_ copies to normalize the data.

### 4.5. Statistical Treatment

GraphPad prism 8.0 software (GraphPad Software Inc., San Diego, CA, USA) was used for statistical analysis. Shapiro–Wilk normality tests were used to determine normal distribution of data subjected to ANOVA. A one-way ANOVA test supplemented with Tukey’s post hoc test was used to compare leaf weight and root length. A two-way ANOVA test supplemented with Tukey’s post hoc test was used for the evaluation of statistically significant differences in the MDA and Fe^2+^ levels obtained from the climatic chamber trial, as well as in the number and length of shoots. This analysis considered two independent variables: the application method and the number of treatment applications. Similarly, for the statistical analysis of results from the olive orchard trial, which included MDA, Fe^2+^ and Proline content, fruit parameters and RT-qPCR results, a two-way ANOVA test supplemented with Tukey’s post hoc test was also employed, considering treatment and orchard location as independent variables. Differences were considered statistically significant when *p* < 0.05.

## 5. Conclusions

This study evaluated the biostimulant activity and protection against Verticillium wilt provided by an onion extract in olive crops, under both controlled conditions and field trials. In climatic chamber experiments, the treatment led to a significant reduction in malondialdehyde (MDA) levels. Additionally, the treatment positively affected root development. Field trials with 4-year-old olive trees under abiotic stress demonstrated that the onion extract promoted a significant increase in budbreak and shoot length, particularly in plants treated by irrigation. Finally, in two commercial orchards naturally infected with *V. dahliae* (Linares and Santaella), the onion extract formulation reduced MDA content in both locations, indicating a potential protection against pathogen-induced oxidative stress. Although the infection rate did not differ significantly between treated and control groups, the log_10_ gene copies of *V. dahliae* were significantly lower in treated trees in both locations, suggesting a reduction in pathogen density. Furthermore, fruit analysis revealed an increase in fat content in the fruits from treated trees. 

For field practitioners, these findings suggest several practical applications. The onion extract should be applied to olive trees through irrigation in consecutive applications to maximize its effectiveness. This method has proven to be more effective in promoting olive tree development and overall health. By incorporating onion extract into integrated pest management strategies, oxidative stress and pathogen density, particularly in orchards affected by *V. dahliae*, could be reduced, while simultaneously increasing the fat content in the fruit.

In conclusion, onion extract standardized in organosulfur compounds demonstrates promising potential as both a biostimulant and a disease management tool for olive crops. Its ability to reduce oxidative stress and improve budbreak and root growth suggests that this extract could help mitigate stress, enhance recovery in affected plants, and control *V. dahliae* infection. These findings highlight the value of onion extract as a sustainable alternative to conventional chemical products and help pave the way for its use in producing more resilient crops. Future research should delve deeper into the underlying mechanisms driving these effects and work towards optimizing application methods to enhance benefits across diverse agricultural contexts.

## Figures and Tables

**Figure 1 plants-13-02499-f001:**
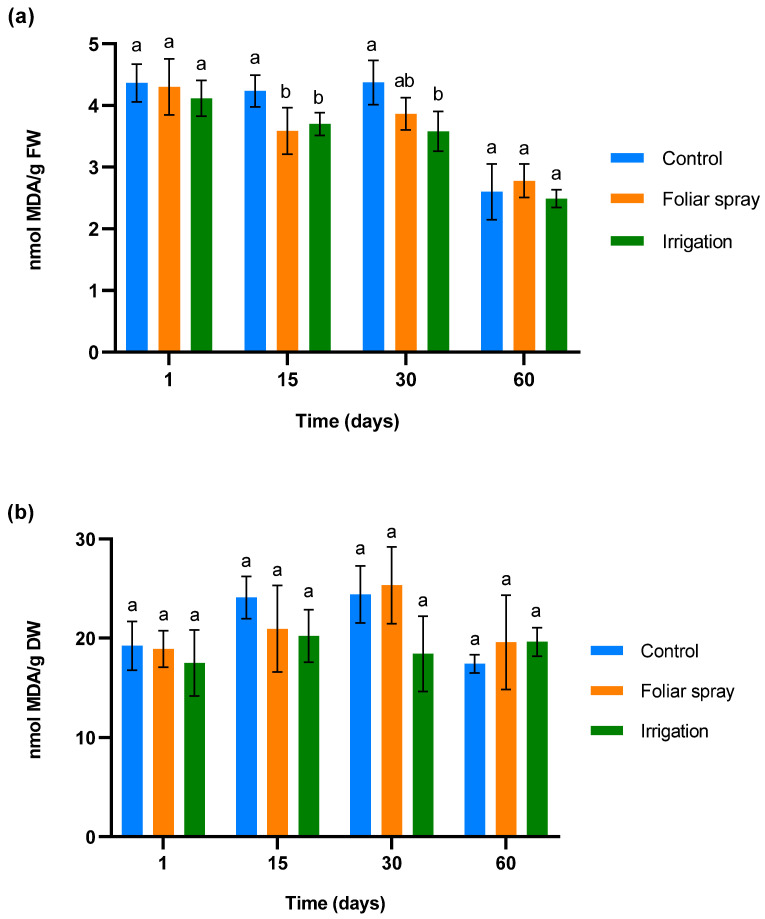
Effect of onion extract formulation applied by foliar spray and irrigation on MDA content of 1-year-old olive plantlets compared to the control: (**a**) leaves (**b**) lyophilized roots. Data are mean ± SD. Bars with different letters indicate significant differences according to the Tukey test (*p* < 0.05).

**Figure 2 plants-13-02499-f002:**
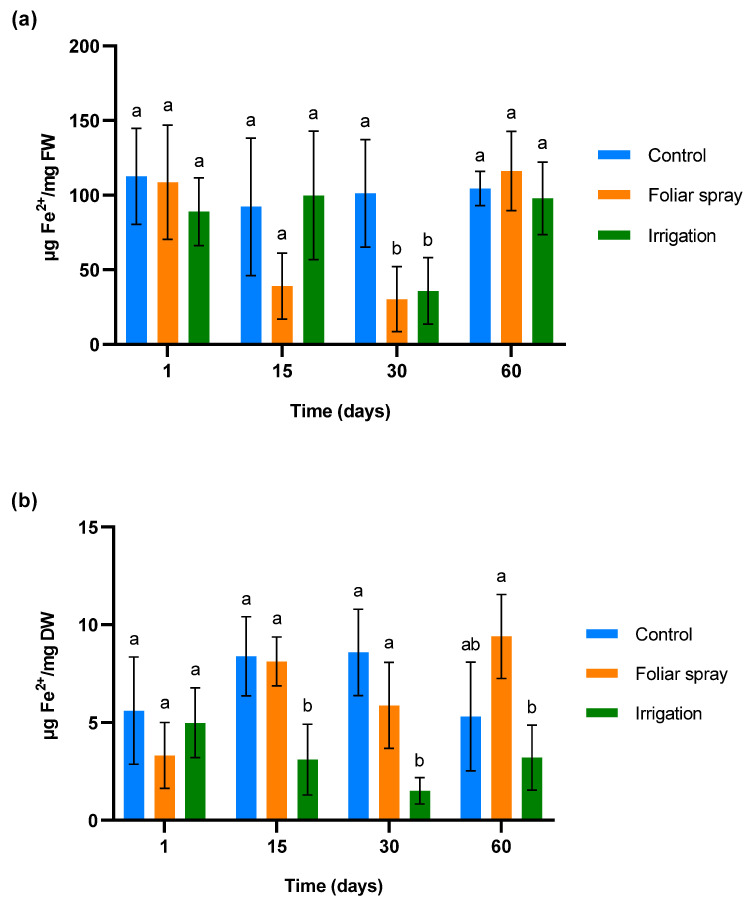
Effect of onion extract formulation applied by foliar spray and irrigation on ferric-reducing power in 1-year-old olive plantlets compared to the control: (**a**) leaves (**b**) lyophilized roots. Data are mean ± SD. Bars with different letters indicate significant differences according to the Tukey test (*p* < 0.05).

**Figure 3 plants-13-02499-f003:**
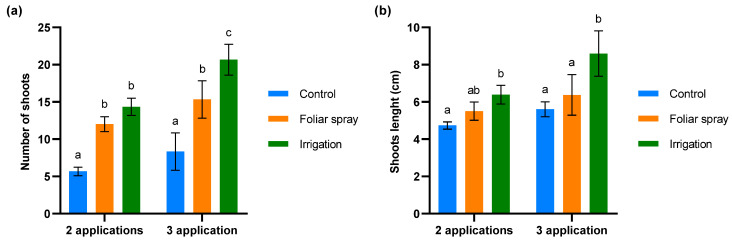
Evaluation of budbreak in 4-year-old olive trees treated with onion extract by foliar spray and irrigation compared to the control: (**a**) number of new shoots (**b**) shoots length (cm). For both panels, bars with different letters indicate significant differences according to the Tukey test (*p* < 0.05).

**Figure 4 plants-13-02499-f004:**
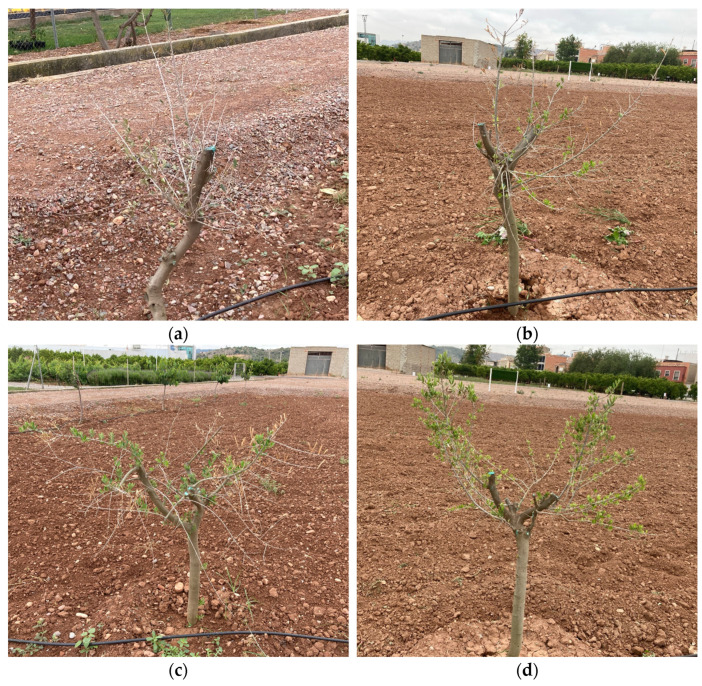
Status of olive trees: (**a**) before the first application (**b**) control plants 20 days after the third application (**c**) plants treated by foliar spray 20 days after the third application (**d**) plants treated by irrigation 20 days after the third application.

**Figure 5 plants-13-02499-f005:**
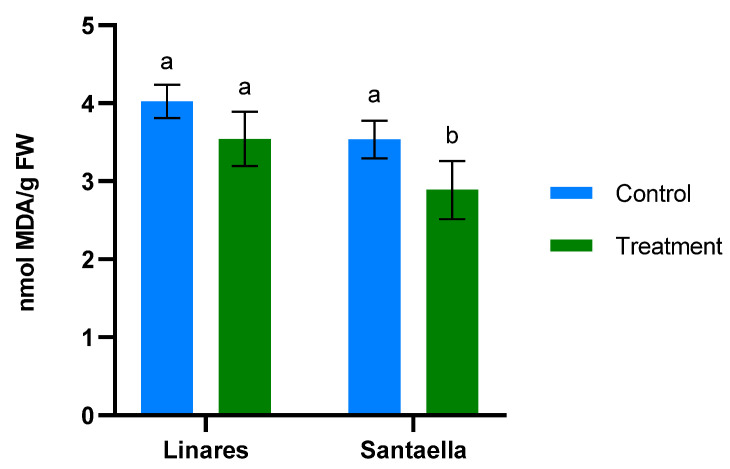
Effect of onion extract formulation applied by irrigation on MDA content of adult olive trees in orchards located in Linares and Santaella. Data are mean ± SD. Bars with different letters indicate significant differences according to the Tukey test (*p* < 0.05).

**Figure 6 plants-13-02499-f006:**
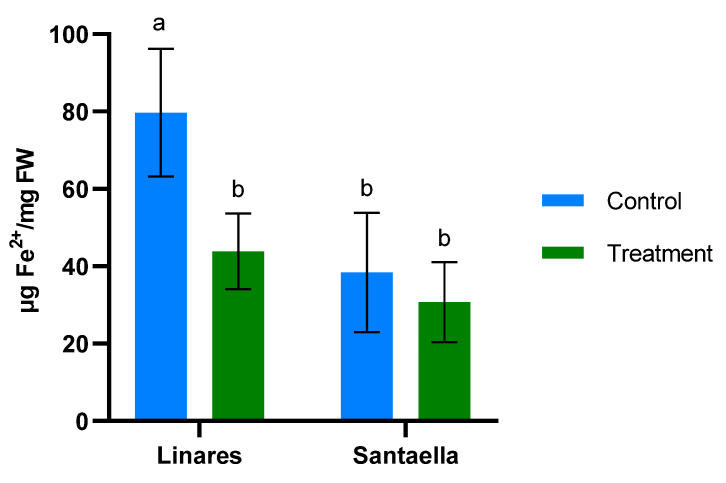
Effect of onion extract formulation applied by irrigation on ferric-reducing power in adult olive trees in orchards located in Linares and Santaella. Data are mean ± SD. Bars with different letters indicate significant differences according to the Tukey test (*p* < 0.05).

**Figure 7 plants-13-02499-f007:**
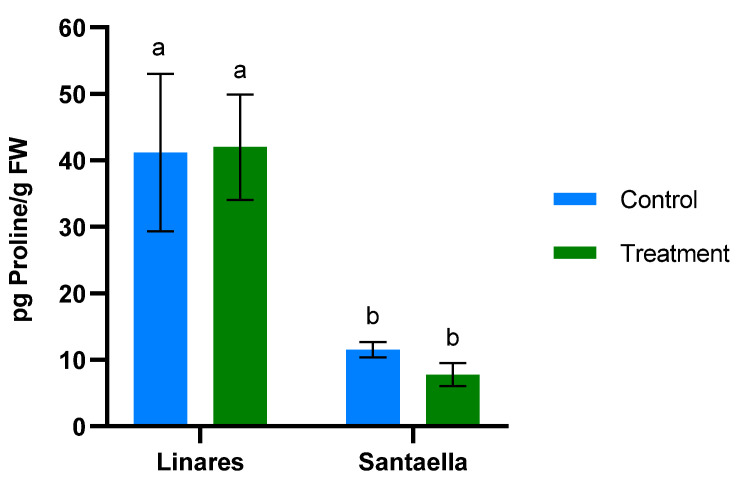
Effect of onion extract formulation applied by irrigation on proline content power in adult olive trees in orchards located in Linares and Santaella. Data are mean ± SD. Bars with different letters indicate significant differences according to the Tukey test (*p* < 0.05).

**Figure 8 plants-13-02499-f008:**
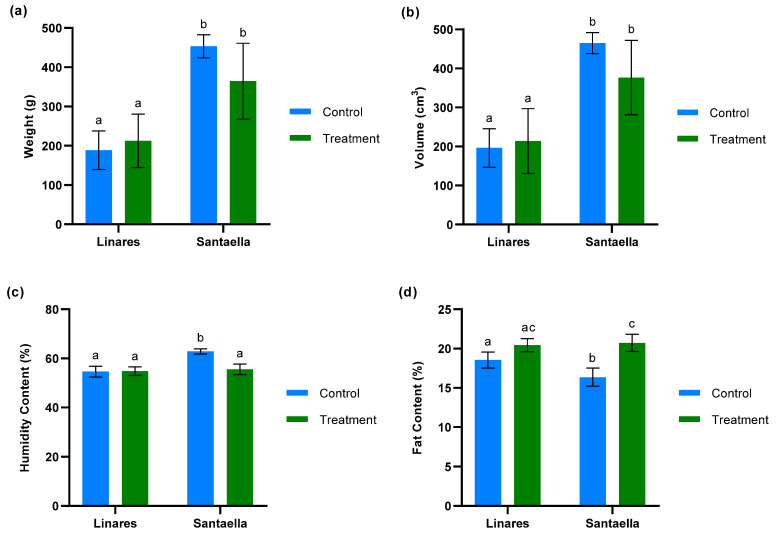
Analysis of fruits from treated and control olive trees in orchards located in Linares and Santaella: (**a**) weight/100 fruits (g) (**b**) volume/100 fruits (cm^3^) (**c**) humidity content/100 fruits (%) (**d**) fat content/100 fruits (%). For both panels, bars with different letters indicate significant differences according to the Tukey test (*p* < 0.05).

**Figure 9 plants-13-02499-f009:**
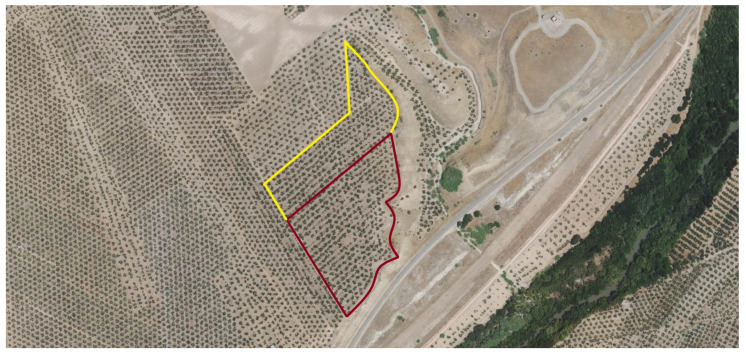
Farm located in Linares, Jaen. The treatment area is outlined in red, and the control area in yellow. Image obtained through the Geographic Information System for Agricultural Plots (SIGPAC).

**Figure 10 plants-13-02499-f010:**
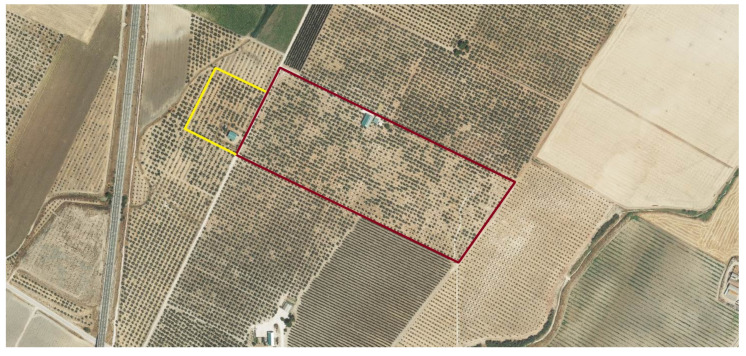
Farm located in Santaella, Córdoba. The treatment area is outlined in red, and the control area in yellow. Image obtained through SIGPAC.

**Table 1 plants-13-02499-t001:** Average leaf weight and root length of olive plantlets treated with onion extract by foliar spray and irrigation under controlled conditions. Different letters indicate significant differences according to the Tukey test (*p* < 0.05).

	Leaf Weight (mg)	Root Length (cm)
Control	131.6 ± 36.1 a	12.2 ± 2.6 a
Foliar spray	139.0 ± 1.4 a	18.6 ± 1.0 b
Irrigation	129.4 ± 12.9 a	19.5 ± 1.6 b

**Table 2 plants-13-02499-t002:** Percentage of infected olive trees and number of gene copies expressed as Log_10_ of *V. dahliae* in control and treated areas in Linares and Santaella orchards. Different letters indicate significant differences according to the Tukey test (*p* < 0.05).

	Infection Rate (%)	Log_10_ Copies
Linares Control	63 ab	4.0 ± 0.3 a
Linares Treatment	70 a	1.4 ± 0.1 b
Santaella Control	60 ab	4.3 ± 0.1 a
Santaella Treatment	53 b	2.0 ± 0.1 b

**Table 3 plants-13-02499-t003:** Treatment applications, evaluations, and sampling dates.

Date	Application	Sampling
April 2021	1st	
May 2021	2nd	
June 2021	3rd	
November 2021		Fruits
April 2022	4th	
May 2022	5th	
June 2022	6th	Leaves

## Data Availability

The original contributions presented in the study are included in the article; further inquiries can be directed to the corresponding authors.

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
