# Peer review of "Evaluation of the Biostimulant Activity and Verticillium Wilt Protection of an Onion Extract in Olive Crops (Olea europaea)"

_plants, 2024, doi:10.3390/plants13172499_

Round 1

Reviewer 1 Report

Comments and Suggestions for Authors

Dear Authors,

here are some notes on the manuscript:

The manuscript fits indubitably the aims and goals of Plants. It is an original work interesting not only specialists, but also agrifood and life science researchers. This study, demonstrating the potential of onion extract as both a biostimulant and an antifungal agent, support its use as a sustainable alternative to chemical treatments, improving olive crop resilience to stress and pathogens.

The article is well put in the context of existing research on the subject. Notwithstanding, some minor revisions are required.                                      The Introduction section, need to add an overview on the active components of extracts from the Allium genus. In fact, it is not enough to say that they "are rich in organosulfur compounds, flavonoids, and other secondary metabolites". 

In the discussion section the authors should illustrate possible hypotheses of attribution of antifungal and/or biostimulant activity to a particular components of the extract (phenols vs organosulfur compound, other bioactive compounds).

Moreover, the effect of the onion extract on corresponding  olive oil quality should be verified or at least hypothesized since the Picual olive cultivars is an olive oil cultivar. 

Author Response

Dear Authors,

Here are some notes on the manuscript:

The manuscript fits indubitably the aims and goals of Plants. It is an original work interesting not only specialists, but also agrifood and life science researchers. This study, demonstrating the potential of onion extract as both a biostimulant and an antifungal agent, support its use as a sustainable alternative to chemical treatments, improving olive crop resilience to stress and pathogens.

The article is well put in the context of existing research on the subject. Notwithstanding, some minor revisions are required.

Dear reviewer,

Thank you for your positive feedback and the time you have dedicated to read and revise our manuscript. We are delighted that you find our work to be a valuable contribution to the field and relevant to a wide range of researchers. We appreciate your recognition of the potential impact of our study on sustainable agriculture. We will carefully address the minor revisions you suggested to further improve the manuscript, and we hope that the changes we make will fully address your concerns and suggestions. In the new version of the manuscript, you can see the modifications along with those requested by other reviewers.

The Introduction section, need to add an overview on the active components of extracts from the Allium genus. In fact, it is not enough to say that they "are rich in organosulfur compounds, flavonoids, and other secondary metabolites". 

Thank you for your comment. You are absolutely right that providing more detailed information on the types of bioactive compounds would add value and better contextualize the nature of the extract for the readers. Following your suggestion, we have introduced a new paragraph in the Introduction section, highlighted in red, to elaborate on the active components of extracts from the Allium cepa. We hope this addition enhances the clarity and depth of the manuscript.

In the discussion section the authors should illustrate possible hypotheses of attribution of antifungal and/or biostimulant activity to a particular component of the extract (phenols vs organosulfur compound, other bioactive compounds).

Thank you for your valuable suggestion. In response, we have expanded the discussion section to include a new paragraph that hypothesizes the potential attribution of the observed antifungal and biostimulant activities of the onion extract to specific bioactive compounds within the formulation. We believe this addition will provide a clearer understanding of the mechanisms at play.

Moreover, the effect of the onion extract on corresponding olive oil quality should be verified or at least hypothesized since the Picual olive cultivars is an olive oil cultivar. 

Thank you for your insightful comment. While our study did not specifically delve into how the onion extract might affect the nutritional quality of olive oil, we agree that this is a very interesting aspect worth considering, especially given that the Picual cultivar is primarily used for olive oil production. To highlight this important point, we have included your observation in the discussion section of the manuscript.

Reviewer 2 Report

Comments and Suggestions for Authors

The article titled "Evaluation of the biostimulant activity and Verticillium wilt protection of an onion extract standardized in organosulfur compounds in olive crops (Olea europaea)" presents significant findings regarding the use of onion extract as a biostimulant and antifungal agent in olive cultivation. Here are some line-by-line comments and suggestions for improvement:

1. The title effectively summarizes the study's focus. However, consider simplifying it for clarity, perhaps by removing "standardized in organosulfur compounds."

2. The abstract is comprehensive but could be more concise. Highlight the main findings more explicitly to capture reader interest immediately.
3.  Please clarify the objectives in the introduction 

4. More detail on the statistical methods used for analysis would enhance transparency and reproducibility.

5. The implications of the findings are discussed well, but integrating more comparisons with existing literature could strengthen the argument for the efficacy of onion extract as a biostimulant, i.e., https://doi.org/10.1007/s10725-024-01128-y

6.  Please suggest specific applications for practitioners in the field in conclusion part of the article. 

Comments on the Quality of English Language

The English is fine.

Author Response

The article titled "Evaluation of the biostimulant activity and Verticillium wilt protection of an onion extract standardized in organosulfur compounds in olive crops (Olea europaea)" presents significant findings regarding the use of onion extract as a biostimulant and antifungal agent in olive cultivation. Here are some line-by-line comments and suggestions for improvement.

Thank you very much for your thorough review and constructive feedback on our manuscript. We sincerely appreciate the time and effort you put into providing these comments and suggestions. Your insights are invaluable, and we will carefully incorporate your recommendations to improve the clarity and quality of the manuscript. We are confident that these revisions will strengthen our work, and we are grateful for your contribution to this process. In the new manuscript, the revisions made are marked up using the Track Changes function.

  1. The title effectively summarizes the study's focus. However, consider simplifying it for clarity, perhaps by removing "standardized in organosulfur compounds."

Thank you very much for your thoughtful suggestion regarding the title. We agree that simplifying the title would enhance its clarity, and we have removed the phrase "standardized in organosulfur compounds" as recommended. We appreciate your input and have made the adjustment in the manuscript accordingly.

  1. The abstract is comprehensive but could be more concise. Highlight the main findings more explicitly to capture reader interest immediately.

Thank you for your assessment. We have modified the abstract to make it more concise and to highlight the main findings to better capture reader interest.

  1. Please clarify the objectives in the introduction 

Thank you for your valuable feedback. We agree that the complexity of the study, which spans multiple levels from climatic chamber experiments to field trials under real-field conditions, made it challenging to clearly define the objectives. Following your recommendation, we have rewritten the objectives in the introduction to ensure they are more clearly articulated. We hope the revised version meets your expectations.

  1. More detail on the statistical methods used for analysis would enhance transparency and reproducibility.

Thanks for this suggestion, we have tried to clarify this point by making some modifications, which can be found in section 4.4 Statistical Treatment. We have provided separate details on the statistical methods used to analyse the results from each scenario—climatic chamber, experimental farm, and olive orchard—clearly specifying the independent variables considered in each case.

  1. The implications of the findings are discussed well, but integrating more comparisons with existing literature could strengthen the argument for the efficacy of onion extract as a biostimulant, i.e., https://doi.org/10.1007/s10725-024-01128-y

Thank you for your valuable feedback. We appreciate your suggestion and we have incorporated additional references and comparisons throughout the discussion section, including the suggested source.

  1. Please suggest specific applications for practitioners in the field in conclusion part of the article. 

Thank you very much for your insightful suggestion. We appreciate your focus on practical applications for practitioners. In response, we have introduced a new paragraph in the conclusion section of the article to address this aspect, providing specific recommendations for field applications. We believe this addition enhance the relevance and applicability of our findings.
